# Protocol for Project Fizzyo, an analytic longitudinal observational cohort study of physiotherapy for children and young people with cystic fibrosis, with interrupted time-series design

Emma Raywood [1], Helen Douglas,[1] Kunal Kapoor,[1] Nicole Filipow,[1] Nicky Murray,[2] Rachel O'Connor,[3] Lee Stott,[4] Greg Saul,[5] Tim Kuzhagaliyev,[6] Gwyneth Davies [1], Olga Liakhovich,[4] Tempest Van Schaik [4], Bianca Furtuna,[4] John Booth,[7] Harriet Shannon,[1] Mandy Bryon,[8] Eleanor Main [1]

For numbered affiliations see end of article.

**Correspondence to**
Ms Emma Raywood;
e.raywood@ucl.ac.uk

## ABSTRACT

**Introduction** Daily physiotherapy is believed to mitigate the progression of cystic fibrosis (CF) lung disease. However, physiotherapy airway clearance techniques (ACTs) are burdensome and the evidence guiding practice remains weak. This paper describes the protocol for Project Fizzyo, which uses innovative technology and analysis methods to remotely capture longitudinal daily data from physiotherapy treatments to measure adherence and prospectively evaluate associations with clinical outcomes.

**Methods and analysis** A cohort of 145 children and young people with CF aged 6–16 years were recruited. Each participant will record their usual physiotherapy sessions daily for 16 months, using remote monitoring sensors: (1) a bespoke ACT sensor, inserted into their usual ACT device and (2) a Fitbit Alta HR activity tracker. Real-time breath pressure during ACTs, and heart rate and daily step counts (Fitbit) are synced using specific software applications. An interrupted time-series design will facilitate evaluation of ACT interventions (feedback and ACT-driven gaming). Baseline, mid and endpoint assessments of spirometry, exercise capacity and quality of life and longitudinal clinical record data will also be collected.

This large dataset will be analysed in R using big data analytics approaches. Distinct ACT and physical activity adherence profiles will be identified, using cluster analysis to define groups of individuals based on measured characteristics and any relationships to clinical profiles assessed. Changes in adherence to physiotherapy over time or in relation to ACT interventions will be quantified and evaluated in relation to clinical outcomes.

**Ethics and dissemination** Ethical approval for this study (IRAS: 228625) was granted by the London-Brighton and Sussex NREC (18/LO/1038). Findings will be disseminated via peer-reviewed publications, at conferences and via CF clinical networks. The statistical code will be published in the Fizzyo GitHub repository and the dataset stored in the Great Ormond Street Hospital Digital Research Environment.

### Strengths and limitations of this study

► This research is directly related to four of the James Lind Alliance top 10 research priorities for cystic fibrosis; simplifying treatment burden, evidence for effectiveness of therapies to delay the onset of lung disease in early life, the application of new technology and exercise as a potential replacement for airway clearance techniques (ACTs).

► The longitudinal observational study design and novel use of daily remote monitoring captures detailed objective data from usual physiotherapy routine care and has the potential to provide real-world evidence to guide ACT practice where randomised controlled trials have failed.

► Variability in synchronisation or use of the ACT sensor and Fitbit or intermittent technical failure of either will make the analysis of remotely collected adherence data complex and challenging.

► Budget constraints, patient preference and the requirement to extract raw heart rate and step count data limited the choice of activity tracker which represented a potential compromise between data quality and willingness to wear the tracker.

► The development of a sustainable big data infrastructure including a pipeline for recording, syncing, processing and analysis of remote monitoring and clinical data was integral to this study and is vital for the future use of these technologies.

**Trial registration number** ISRCTN51624752; Pre-results.

## INTRODUCTION

Cystic fibrosis (CF) is a life-limiting inherited condition, which affects over 10 000 people in the UK. Despite recent treatment advances, including modulator therapies that target the underlying cause of the disease,[1] CF remains progressive and incurable. People with CF are

susceptible to repeated respiratory infections due to thick respiratory mucus, which can lead to irreversible lung damage. Daily treatment to slow the progression of lung disease, and therefore the onset of respiratory failure, is the primary aim of almost all current CF therapies.[2] Daily physiotherapy, including airway clearance techniques (ACTs), physical activity and exercise, is believed to mitigate the progression of CF lung disease.[3]

People with CF undertake a median of 10 different concurrent treatments that take an average of 2 hours each day.[4] This high daily treatment burden (and high cost of care) from childhood has become a driver for research to maximise effective care and minimise unnecessary therapies. Reducing treatment burden has been identified as the top CF research priority by the James Lind Alliance.[5] Follow-up research found that although ACTs were perceived as very important, they were considered the most burdensome daily treatment.[4] ACTs are stressful for people with CF and their families, and adherence can be low. Of those questioned, 70% said they regularly missed some of their daily treatments, most commonly avoiding ACTs or nebulised therapies.[4] Another study found those with the lowest adherence to chest physiotherapy had worse lung function, more exacerbations and consequently had higher health costs.[6]

Another of the James Lind Alliance top 10 research priorities for CF is to identify the specific therapies that could delay and prevent the progression of lung disease in early life.[5] Despite over 70 years of ACTs in routine CF clinical practice, the evidence base to guide treatment remains weak. A number of challenges exist for researchers in the field; a plethora of physiotherapy ACTs and devices exist, providing a bewildering choice for therapists and patients, with the long-term effects of different devices, techniques or non-adherence being poorly understood. Traditional research methods (including randomised controlled trials) have failed to produce credible evidence for optimal ACT type, dose, frequency or duration.[7 8] Established solitary outcome measures (eg, forced expiratory volume in 1 second; $FEV_1$) are insensitive to change in mild CF lung disease, and are not useful endpoints for physiotherapy clinical trials. Furthermore blinding is not possible, patient or practitioner preferences can confound results of trials and ethical concerns about the complete removal of ACTs persist.[3] There is evidence that patients with the lowest self-reported physical activity levels have poorer health,[9] but the use of exercise as an ACT remains controversial,[10] despite the fact it is popular with patients.[5]

Advances in technology, including increased use of electronically chipped devices, electronic patient records and the growth of big data analytics, are providing fresh opportunities for physiotherapy research. These may facilitate clarity and certainty about effective therapies and help to reduce treatment burden. Big data techniques, including machine learning and unsupervised clustering, are useful for analysis of healthcare data, which is often complex, unstructured and from multiple sources. However, they are yet to be used to build credible evidence to guide physiotherapy for people with CF.

## Project Fizzyo

This paper describes the protocol for Project Fizzyo, which uses innovative technology to capture detailed longitudinal data from children and young people with CF (CYPwCF) undertaking daily physiotherapy treatments. Real-time breath pressure during ACTs are captured daily, as well as heart rate and step count during physical activity. The interrupted time-series study design facilitates evaluation of interventions such as ACT feedback and ACT-driven gaming.

The aims of Project Fizzyo are (1) to monitor and understand adherence to daily ACT prescriptions, (2) to evaluate associations between ACT adherence and clinical outcomes under different study conditions (including feedback and gaming), (3) to monitor and understand daily patterns of physical activity in relation to published recommendations, and evaluate any associations with clinical outcomes and (4) to use big data analytics to seek out important data signals in relation to optimising ACT and exercise prescriptions for CYPwCF.

## METHODS
### Study design

Project Fizzyo is an analytic longitudinal observational cohort study with an interrupted time-series design. Clinically prescribed ACT or exercise prescriptions for individual participants are continued as normal for the 16 months of the study. Each participant uses two remote monitoring sensors to record data daily. These are: (1) a bespoke Fizzyo ACT breath pressure sensor, inserted into their regular ACT device (Acapella Choice, Aerobika, AstraTech PEP or Pari PEP) and (2) a Fitbit Alta HR activity tracker (Fitbit, San Francisco, USA) to record daily physical activity (heart rate and step count). Participants synchronise (sync) data using software applications (apps) for each sensor, on a tablet computer provided for the study.

ACT feedback (daily number of breaths) and ACT-driven computer gaming are introduced and removed in an interrupted time series (figure 1), via the specially developed Fizzyo app. Baseline ACT patterns are recorded (during months 0–2: standard phase), then the sequential introduction of ACT feedback (during months 2–14: feedback phase) and ACT-driven gaming (during months 4–12: gaming phase) will allow any effect on breathing patterns, adherence or clinical outcomes to be observed. The removal of gaming (at month 12) and feedback (at month 14) will facilitate observation of lasting, temporary or feedback/game dependent behaviour changes.

## Patient and public involvement

This study addresses top research priorities identified by people with CF, their families, carers and clinical teams as part of the James Lind Alliance CF Priority Setting

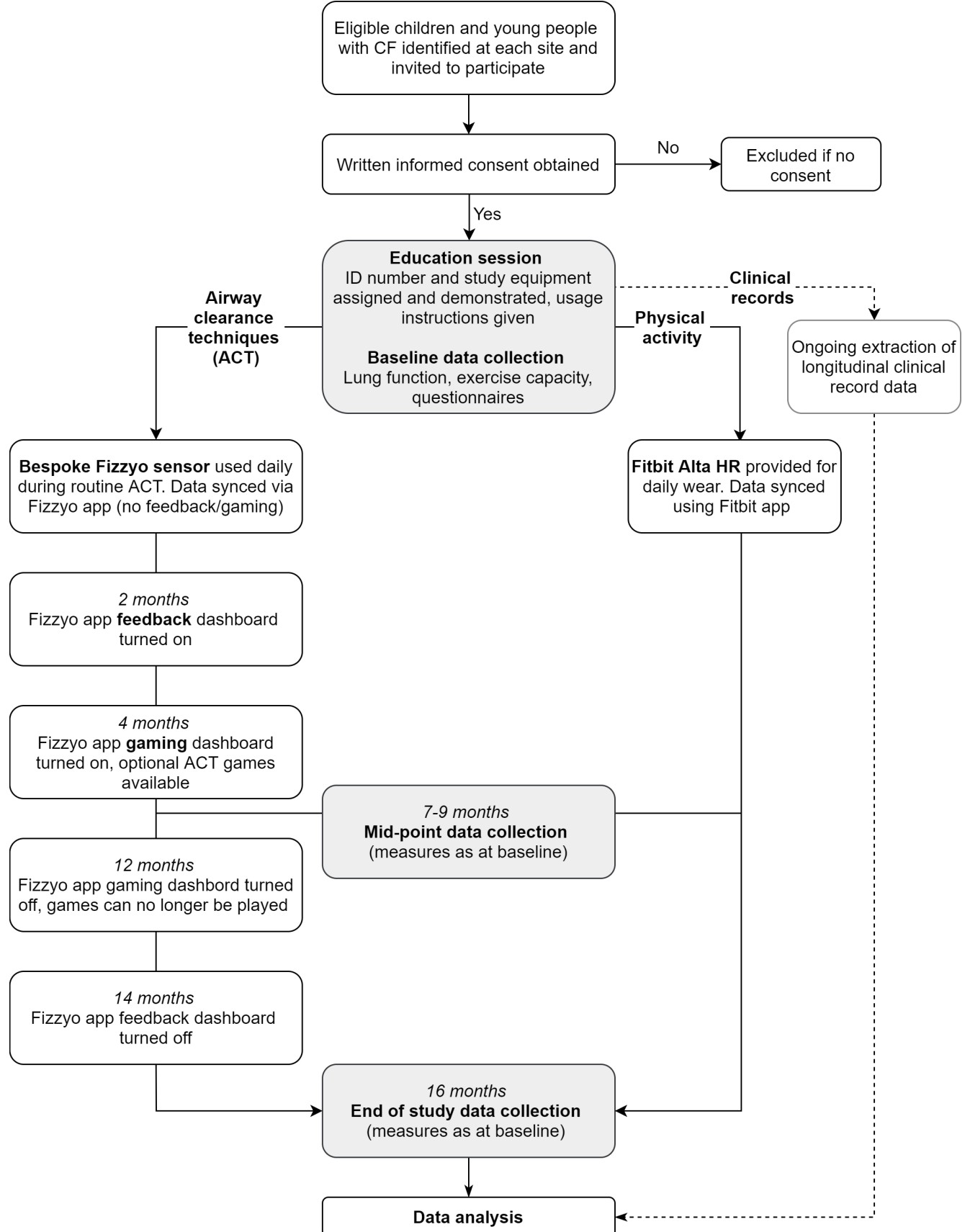

**Figure 1** Study design flowchart of the 16-month study per participant. Participant assessment visits are in shaded boxes. ACTs and physical activity are recorded daily using electronically chipped devices (a bespoke ACT sensor and a Fitbit activity tracker). The specially developed Fizzyo app, as well as syncing ACT data throughout the 16 months, gives participant feedback in months 2–14 and ACT-driven gaming is available in months 4–12. ACT, airway clearance technique; CF, cystic fibrosis.

Partnership.[4] The ACT sensor and gaming were developed as part of the BBC2 programme, 'The Big Life Fix' at the request of a family with two sons who have CF.[11] This idea was developed further by the study team, which includes a parent of children with CF, physiotherapy specialists and product design specialists. Children from Great Ormond Street Hospital (GOSH) and their families advised on the choice of Fitbit, development of the ACT sensor (including the addition of lights to indicate breathing), customisability of the Fizzyo app and games, the study design and information materials. Dissemination of study results to both participants in the study and the wider CF population is planned.

## Participants

To be eligible, participants at recruitment were: (1) aged 6–16 years, (2) diagnosed with CF and under the care of a participating London paediatric CF centre (either GOSH, the Royal London Hospital (RLH) or the Royal Brompton Hospital (RBH), including shared care patients) and (3) prescribed one of the four ACT devices compatible with the ACT sensor, at least once a day as part of routine ACTs.

Patients were excluded if (1) they/their parents did not or could not provide informed consent, (2) they were not prescribed one of the four specific ACT devices as part of their routine daily treatment (pre-study self-reported non-adherence to prescribed ACT did not prevent children from participation), (3) had undergone lung transplantation or (4) had a clinically significant medical condition other than CF.

Children meeting the eligibility criteria were approached by their clinical team with the study information sheets. Participant information sheets affirm that no direct benefits are promised to participants, although they may find the computer games help them to enjoy and engage with airway clearance. After 16 months the study will end for each participant and the ACT sensor and Fitbit will be returned. Participants can choose to keep the devices, with restored games and feedback for a short period at the end of their participation, on the condition that they continue to sync data until the remaining period of data collection is concluded for all participants (December 2020).

Participants were recruited from September 2018 to July 2019. The recruitment window was scheduled to be open for 6 months per site, though for the first and largest recruiting site (GOSH) an extra month was added to allow for a period of pilot data collection to resolve early technical issues. The initial recruitment target of 160 participants was reduced to 145 due to manufacturing delays for the bespoke Fizzyo sensor, which was not a threat to meeting the aims of the study. This study, with 145 CYPwCF is the largest CF population sample in any physiotherapy study undertaken in Europe to date. Daily data from these children over 16 months will provide sufficient precision of estimation in evaluating the signals of adherence to therapy.

The number of children (recruitment period) at each hospital were: at GOSH 75 (10 September 2018 to 16 April 2019), RBH 40 (30 November 2018 to 31 May 2019) and RLH 30 (03 January 2019 to 01 July 2019). Data collection is ongoing and is predicted to end in December 2020.

## Data collection

Informed consent to participate in the study was given by a parent, guardian or participant (if of suitable age and understanding) and assent was granted by younger participants. Each participant received study equipment (table 1, figure 2), to record and synchronise data, and was issued a unique anonymous Microsoft login account with a strong password (managed by the study team) for logging into the tablet, Fitbit app (Fitbit) and bespoke Fizzyo app. A researcher demonstrated how to use the equipment to participants. These instructions and technical support contact details were provided as a printout and webpage link via QR code.

At recruitment, baseline assessments of lung function (spirometry: $FEV_1$, forced vital capacity, forced expiratory flow$_{25–75}$) and exercise capacity (10 metre modified shuttle walk test, with a Polar H10 heart rate monitor and Fitbit) were undertaken. A quality of life questionnaire (Revised CF Questionnaire; CFQ-R) and a physiotherapy questionnaire was completed by participants with a member of the study team digitally via REDCap. Spirometry is always performed prior to exercise capacity assessment or after at least 1 hour of rest following maximal exertion. All tests are carried out in accordance with the appropriate guidelines. These measurements were repeated midway (7–9 months) and will be collected again at the end of the study (16 months, estimated completed in December 2020). Assessments are carried out on days when participants have a clinic appointment or routine hospital admission (but not during an exacerbation).

Data are pseudonymised, stored and analysed in the secure GOSH Digital Research Environment (DRE, Aridhia, Edinburgh, UK). The DRE is a cloud-enabled, dedicated GOSH secondary use datastore of medical data from electronic patient records, remote monitoring sensors and external databases. Embedded analytic tools will enable approved researchers to perform statistical analysis within the DRE. Data encryption and transfer protocols and data sharing and processing agreements ensure data are processed in a responsible, lawful, secure and confidential way presented with NHS information governance standards, the UK data protection act (2018) and the EU GDPR (2018).

## Remote monitoring
### Activity tracker

A Fitbit activity tracker was chosen as Fitbit allow developers to use an application programming interface (API) for extraction of processed granular heart rate and step count data. The specific activity tracker model, the Fitbit Alta HR (released in 2017), was chosen with input from CYPwCF and their families at GOSH, from a range of

**Table 1** Study equipment for participants

| | ACT sensor | Activity tracker | Tablet computer |
|---|---|---|---|
| **Description** | Bespoke Fizzyo Sensor. Wireless battery powered pressure sensor that attaches to ACT devices to measure pressure during ACT. | Fitbit Alta HR. Wrist-worn activity tracker with heart rate sensor to record daily physical activity. | Linx 12X64 Windows 10 tablet computer. Required to host remote monitoring apps and games. |
| **Features** | ▶ Microelectromechanical system-based piezoresistive sensor. ▶ Two buttons for power on/off and game control input. ▶ 1 MB flash-based storage for approximately 227 hours of data. ▶ 7 days use from full charge. To fully charge: 70 min. | ▶ Customisable display with clock face. ▶ Photoplethysmographic PurePulse technology heart rate sensor. ▶ 3-axis accelerometer movement detection. ▶ Memory capacity for 7 days of full data. ▶ 7 days use from full charge. To fully charge: 1–2 hours. | ▶ Intel Atom x5 processor. ▶ 4GB RAM. ▶ Bluetooth 4.0. ▶ Wi-Fi 802.11. ▶ 64GB memory storage. ▶ 5–7 hours use from full charge. To fully charge: 3 hours. |
| **Data collected** | ▶ Time-stamped ACT pressure data (10 Hz). | ▶ Heart rate (variable sampling frequency; 6–30 times/min), ▶ Step count (per min). | ▶ Extraction and transmission of Fizzyo sensor and Fitbit data via sync with apps. ▶ Gaming data. |
| **App Details** | Fizzyo app (FizzyoHub) ▶ Developed using Visual Studio for Windows 10. ▶ Bluetooth syncing of the Fizzyo ACT sensor. – *Standard phase*, syncing only. – *Feedback phase*, daily and monthly breath count graphs enabled. – *Gaming phase,* games (developed in Unity, hosted in Microsoft store) able to be installed and played. | Fitbit app (for Windows 10) ▶ Developed by Fitbit Inc. ▶ Bluetooth syncing of the Alta HR. ▶ Patient facing dashboard displaying daily and historical graphs of step and activity patterns. ▶ Feedback on progress against daily and longer-term goals. | Windows store app ▶ Required to install and update Fitbit app and Fizzyo app and ACT games. |

ACT, airway clearance technique; app, Computer application.

Fitbit devices. It is an appropriate size for the wrists of children from 6 years old and is simple to use.

The Fitbit has a photoplethysmographic heart rate sensor for heart rate measurement, which is essential to physiologically quantify adherence to recommended minutes in daily moderate to vigorous physical activity (MVPA). Heart rate data are collected continuously with a variable sampling frequency (table 1) whenever the Fitbit is worn during the 16 months. Participants are asked to wear the Fitbit during all waking hours except when bathing/swimming (it is not waterproof), and to sync data daily via Bluetooth using the Fitbit app, or at a minimum once per week: the memory capacity of the tracker. Participants are asked to charge the battery at least once per week or when notified by the device/app.

The standard Fitbit app provides feedback data, which is available for participants to view throughout the study. After participants sync their device, the study team extract anonymised data from the Fitbit cloud using the API. It

is downloaded to the Fizzyo data cloud and then transferred securely into the DRE (figure 2).

### ACT sensor
A bespoke wireless sensor was developed for Project Fizzyo with involvement from CYPwCF and their families at GOSH, physiotherapy specialists and product design specialists. The sensor is battery powered and clips on to compatible ACT devices (figure 3) in the same way that pressure manometers are clipped in during ACT training or use. It measures air pressure changes inside the physiotherapy ACT device during breathing.

Participants are instructed to use the sensor to record all ACT treatments during the study. To reduce the chance of data loss, participants are asked to sync data once per day using the Fizzyo app and charge the sensor at least once per week. Pressure data are stored on the sensor's encrypted memory chip (capacity 1 Mb) and can be synchronised either immediately after treatment or

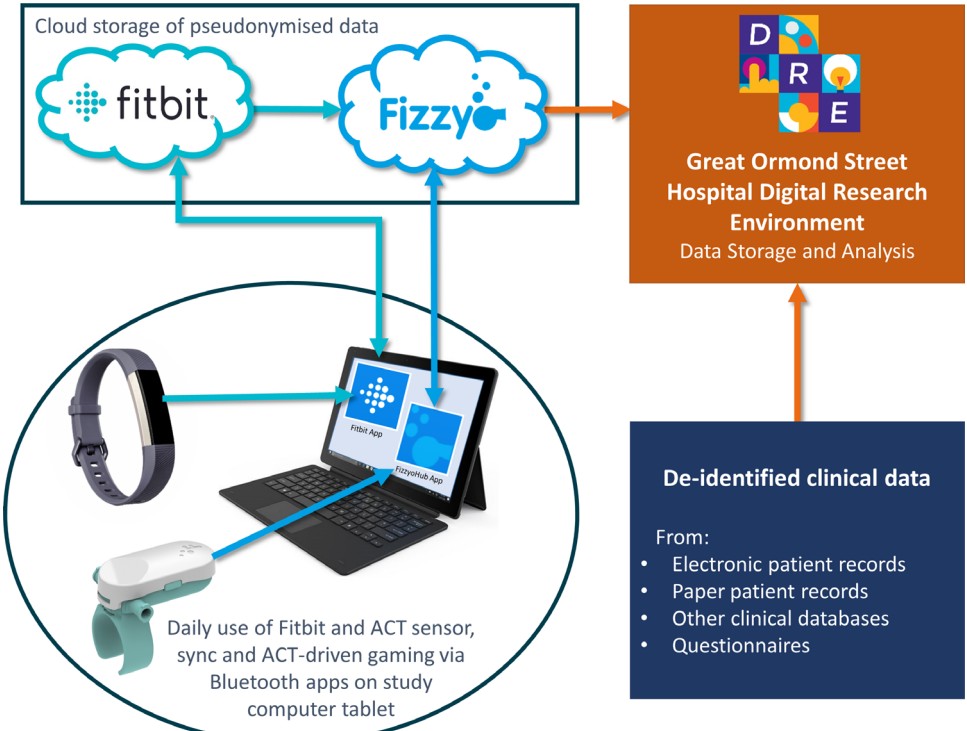

**Figure 2** Project Fizzyo data collection pathway. Remote monitoring sensors (Fizzyo sensor, Fitbit) connect via Bluetooth to sync data with device-specific apps on a study tablet. Anonymous data is sent to the Fizzyo cloud (either directly or via the application programming interface from the Fitbit cloud) and then linked with de-identified clinical records in the Great Ormond Street Hospital Digital Research Environment. ACT, airway clearance technique.

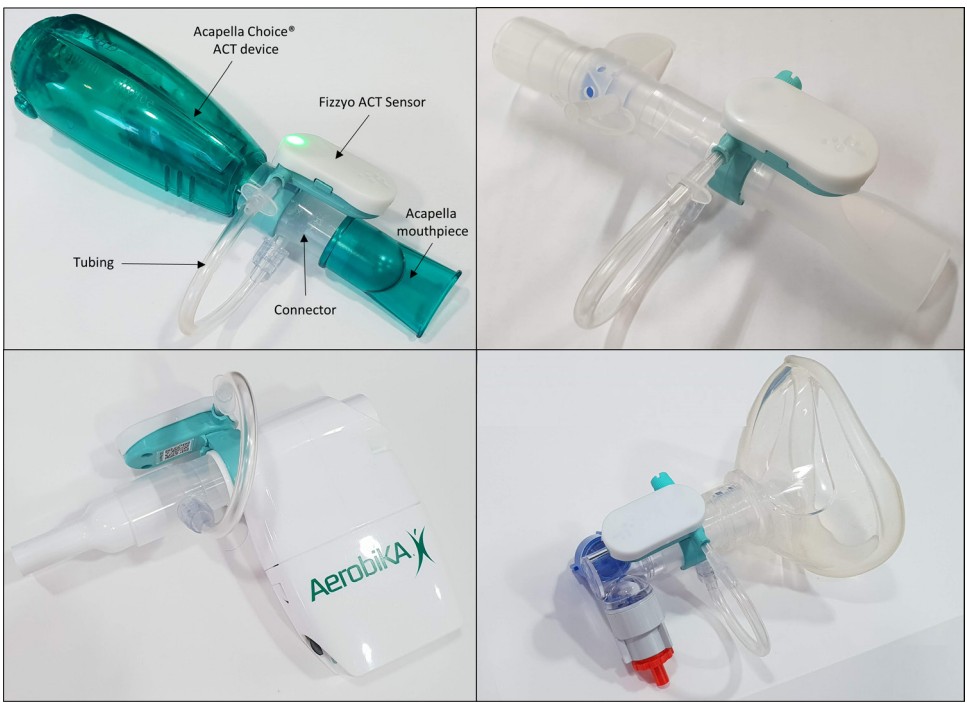

**Figure 3** Fizzyo sensor and connectors attached to four airway clearance devices. Left are oscillatory PEP devices: an Acapella choice (labelled) and Aerobika. Right shows non-oscillatory PEP devices the Pari PEP (top) and AstraTech PEP (with a mask). Breath pressure changes within the airway clearance device are recorded by the sensor. PEP, positive expiratory pressure. ACT, airway clearance technique.

stored to sync at a later time. Immediately upon syncing data to the Fizzyo cloud (figure 2) basic breath count processing occurs, to provide breath count feedback to participants within a few seconds (during the feedback phase of the study only). Detailed breath count analysis will be performed on cleaned pseudonymised data in the DRE.

### ACT-driven gaming

Windows-compatible games have been developed using a bespoke structured gaming framework (https://github.com/Fizzyo/FizzyoFramework-Unity) in Unity (Unity Technologies, San Francisco, USA), through collaboration with Microsoft, UCL and Abertay University students and CYPwCF at GOSH. The framework takes real-time inputs during ACT treatments (expiratory pressure during breathing and a button press) from the Fizzyo sensor, and uses these to simulate standard joystick inputs.

Five bespoke games with different mechanics are available to participants (including Qubi, Archipelayo and Egg Toss); including gamification features such as anonymous high scoreboards and achievements. Games were designed to encourage the player to blow in the prescribed breath length and pressure range (slightly extended expiration, 15–20 cmH$_2$O), typically in multiple cycles of 8–10 breath sets with pauses for huff/cough. The number of breaths and cycles in the game can be altered by the player, reflecting the personalised ACT prescriptions of children in the study. Gaming is optional for participants during ACTs, but whether games are played or not, can be identified, including which games are played during specific sessions.

### Participant questionnaires

Physiotherapy prescription including: ACT device(s), number of breaths and sets per day, estimated duration of ACT, exercise prescription and self-reported routine physical activities for individuals are collected by participant questionnaires at recruitment, mid-point and study end. An age-specific CF questionnaire (CFQ-R: 6–11 years, 12–13 years or 14 years+) is also administered at these timepoints for all participants. A parental CFQ-R is completed for children under 14 years of age. The CFQ-R is a CF-specific measure of quality of life, with domains including respiratory and gastrointestinal symptoms, treatment burden and daily functioning; it is currently the best validated CF-specific patient-reported outcome measure.[12]

### Clinical data

GOSH clinical data are extracted from electronic patient records (EPIC Systems, Wisconsin, USA), pseudonymised and uploaded directly to the DRE. Historical data from older patient record systems (that are not available in electronic patient records) will be extracted, pseudonymised and uploaded to the DRE separately. Where possible, this dataset will contain a participant's entire clinical record including physiotherapy and known key features for CF recorded in the UK CF Registry,[13] such as genotype,

anthropometry, medications, hospital admissions, lung function, pancreatic status, microbiology, co-morbidities, ACT prescription (including documented changes to technique) and exercise test results.

At RLH and RBH electronic patient records are not currently available and clinical information, including demographics and ACT prescription will be extracted from clinical records and databases. Data will be pseudonymised at source, and transferred by secure file transfer protocol to the DRE.

A clinical data processing pipeline will be developed to summarise features describing patient demographics and clinical status as frequencies (eg, number of hospital admissions) or rates (eg, number of hospital admissions per year).

### Data analysis
#### Adherence to physiotherapy

The primary outcome measures are: adherence to a participant's daily ACT prescription (during different study phases: standard, feedback, gaming), and adherence to daily physical activity recommendations. ACT sensor (breath pressure) and Fitbit (heart rate, step count) data are processed via a data pipeline in R,[14] within the DRE to describe these outcomes. Clinical data outcomes will be used to investigate any relationship between physiotherapy adherence and specific clinical profiles.

Adherence to medical treatment is the extent to which a patient follows the recommendations agreed upon with health professionals.[15] As physiotherapists give personalised ACT prescriptions of varying breath count, set count and treatment session number per day, personalised adherence scores will be calculated. Daily ACT adherence will be quantified by the number and type of breaths recorded in a treatment session and per day. Cumulative, aggregate and weekly average scores will also be calculated. Adherence to the prescribed breath count (proportion of completed breaths against prescribed breaths per treatment; usually around 100 breaths), prescribed breath pressures (proportion of ideal breath pressure breaths (15–20 cmH$_2$O) per treatment) and prescribed breath length (proportion of expiratory breaths at ideal length of at least 1–2 sec) and also the presence of gaps for huff and cough between sets will be assessed. A composite adherence score based on these parameters combined per session, per day and per week will be calculated.

WHO guidelines recommend that all *children and youth aged 5–17 years should accumulate at least 60 min of MVPA daily*.[16] Adherence will be measured against this target for all participants. Time in MVPA is calculated daily as time where heart rate is greater than a personalised threshold value (usually around 120 beats/min), which will consider an individual's resting heart rate, and maximal heart rate from the exercise test. A number of published approaches,[17] are being investigated to determine the optimal MVPA heart rate threshold as no clear approach currently exists in the literature for the study

**Table 2** Data pipeline stages and features for cluster analysis

| Sensor type of data | Data pipeline step | | Featurisation of variables for cluster analysis | |
|---|---|---|---|---|
| | Cleaning | Labelling (of) | Descriptive | Describing adherence* |
| **ACT Sensor** Pressure–Time | ▲ Removal of blank, duplicate and non-physiological values.<br>▲ Any non-linear baseline drift corrected, using a sparsity based de-noising approach.[19] | ▲ Treatment sessions<br>▲ Pressure peaks<br>▲ Breaths<br>▲ Breaks between breaths<br>▲ Sets of breaths | ▲ If any breaths recorded on a day Y/N<br>▲ Breath count†<br>▲ Breath length†<br>▲ Breath peak pressure†<br>▲ Treatment duration†<br>▲ Number of treatments per day†<br>▲ Number of sets in a treatment†<br>▲ Number of breaths per set† | ▲ Adherence score (proportion of days with any breaths recorded per total number of days)<br>▲ Breath count adherence (proportion of completed breaths against prescribed breaths per treatment)<br>▲ Set adherence (proportion of sets against prescribed sets per treatment)<br>▲ Treatment session adherence (proportion of completed treatments against prescribed treatments)<br>▲ Pressure adherence (proportion of ideal expiratory pressure breaths per treatment)<br>▲ Breath length adherence (proportion of expiratory breaths at ideal length per treatment) |
| **Fitbit** Heart rate and step count | ▲ Removal of erroneous or non-physiological data (caused by improper wearing, depleted battery, full memory capacity due to infrequent syncing).<br>▲ Heart rate sampling frequency made consistent (per min) using a rolling average | ▲ Gaps in data<br>▲ Wear time (from heart rate data)<br>▲ Awake wear time<br>▲ Time in MVPA using personalised heart rate cut-off value[17]<br>▲ Points crossing MVPA cut-off value | ▲ Heart rate<br>– Resting[20]<br>– Peak<br>– Density and variability<br>– MVPA threshold switches<br>▲ Step count<br>– Daily step count<br>– Density and variability<br>– Active minutes (greater than a threshold value)<br>▲ Combined<br>– Active minutes both heart rate and step count.<br>– Step count during periods of MVPA | ▲ Daily time in MVPA compared with 60 daily minutes recommended.<br>▲ Weekly time in MVPA, 7 day window compared with 420 weekly minutes recommended. |

Not all features shown.
*Features quantified as adherence per prescribed treatment session, per day, per rolling 7 day week, or other time point as required for analysis. ACT prescription is taken from clinical records and physiotherapy questionnaires.
†Total and/or average per treatment, SD, min, max values.
ACT, airway clearance technique; MPVA, moderate to vigorous physical activity.

population. Step count data will be used to support heart rate data and assist with identification of sedentary time (for resting heart rate calculation) but will not be the primary method for activity estimation as some activities do not generate a high step frequency despite being vigorous activities (eg, cycling).

### Data processing pipeline

A data processing pipeline enabling large amounts of remotely monitored sensor data to be processed efficiently, reliably and reproducibly was developed in collaboration with Microsoft computer and data science engineers. It was developed using preliminary data collected in the first 3 months of data collection (September to December 2018) and then tested on a larger dataset (to April 2019).

The pipeline ensures data are processed consistently and data validation checks at each stage alert to errors. The pipeline is being refined throughout the ongoing data collection period but, briefly, it processes data through three main steps: *data cleaning* to remove errors, *data labelling* to mark and measure predefined constructs from raw data (eg, distinct breaths from pressure traces) and finally *data featurisation*, the quantification of variables for cluster analysis. These steps and the main features for analysis are summarised in table 2.

### Statistical analysis

The large amount of heterogeneous data to be recorded and collected for each participant will be analysed using R and visualised using R Shiny apps. Any changes in adherence to physiotherapy prescriptions or recommendations over time or in relation to ACT gaming or feedback, will be quantified. As CF phenotypes and physiotherapy adherence are complex and multifactorial, cluster analysis will define groups of individuals based on measured characteristics to identify subgroups of participants with distinct physical activity and/or ACT adherence profiles.

The Fitbit and ACT sensor dataset currently contains 50+ variables (key features shown in table 2), with more to be added from ongoing clinical record extraction. A correlation analysis of the features describing physiotherapy behaviours will remove features which are highly correlated. Variables with a Gaussian distribution will be normalised. Dimensionality reduction will be performed via principal component analysis. These methods will identify the most relevant and independent variables for cluster analysis to ensure a robust definition and visualisation of clusters.

Multiple features of ACT treatment sessions or activity days identified from the principal component and correlation analyses will be grouped using unsupervised cluster analysis, such that both the within-cluster similarity and between-cluster dissimilarity are strong. Unsupervised cluster analysis with the k-means method[18] is a popular clustering technique and involves grouping multifactorial data into a pre-specified number of clusters. Numerous rapid computations comparing the relationship between multiple variables will determine the optimal cluster

centres with the best separation between groups. The silhouette metric will be used to assess cluster homogeneity and separation, to identify the optimal number of clusters.

The associations, patterns and trends of individual physiotherapy adherence clusters to health outcomes (eg, lung function, number of hospital admissions, infections, antibiotics, etc) will be investigated with data from clinical records. Individual participant changes between clusters over time, or in relation to gaming or feedback, will also be investigated.

## ETHICS AND DISSEMINATION

Ethical approval for this study (IRAS ref: 228625) was granted by the London-Brighton and Sussex, NREC (18/LO/1038). This study is also registered with the GOSH/UCL GOS ICH joint research and development office (ref: 17IA06) and has been adopted into the NIHR Clinical Research Network. Study findings will be disseminated via peer-reviewed publications in open-source journals, relevant national and international conferences and via CF clinical and patient networks. Results and the statistical code will be published in the Fizzyo GitHub repository and the dataset stored and available through the GOSH DRE. Collaboration with Microsoft has produced two high-quality videos already available on YouTube and the production of further high-quality resources to disseminate results to both the CF population and participants in the study is planned.

**Author affiliations**
[1]Physiotherapy, Respiratory, Critical Care and Anaesthesia Section, UCL Great Ormond Street Institute of Child Health, London, UK
[2]Paediatric Cystic Fibrosis Unit, Royal Brompton and Harefield NHS Foundation Trust, London, UK
[3]Paediatric Cystic Fibrosis Centre, Royal London Hospital, Barts Health NHS Trust, London, UK
[4]Commercial Software Engineering, Microsoft UK Ltd - Reading, Reading, UK
[5]Microsoft Research Lab, Microsoft Research Ltd, Cambridge, UK
[6]Computer Science Department, UCL, London, UK
[7]Digital Research, Informatics and Virtual Environments (DRIVE) Unit, Great Ormond Street Hospital for Children NHS Foundation Trust, London, UK
[8]Department of Paediatric Psychology, Great Ormond Street Hospital for Children NHS Foundation Trust, London, UK

**Acknowledgements** With thanks to all participants and their families, the clinical CF Teams at each of the recruiting sites, the GOSH DRE team and Microsoft UK, especially Haiyan Zhang, Simon Jackson, and Microsoft CSE including Josh Lane, Pete Roden, Stephanie Marker, Christian Robles, Kristjana Popovski, Hannah Kennedy and Kristin Ottofy. Also Ryan White, Michael Woollard and Alan Bannon for electronic engineering input, Dean Mohamedally and UCL students (computer science), and Jamie Bankhead and the team at Abertay University for gaming designs and UCL physiotherapy MSc students. We thank Trudell Medical for an ACTNow education award (October 2019).

**Collaborators** Haiyan Zhang; Simon Jackson; Josh Lane; Pete Roden; Stephanie Marker; Christian Robles; Kristjana Popovski; Hannah Kennedy; Kristin Ottofy; Ryan White; Michael Woollard; Alan Bannon; Dean Mohamedally; Jamie Bankhead.

**Contributors** EM, MB, HD, HS, GS and LS conceived and designed the study. GS perceived and developed the sensor and with LS and TK the data processing infrastructure including the bespoke app. EM, ER and HD developed the protocol

with input from all investigators. ER, HD, NM and RO'C recruited and support all participants. JB is the DRE data steward and facilitator of electronic patient record extraction. KK, NF, OL, TVS and BF developed the data processing pipeline and preliminary analysis with EM, GD, NM, RO'C, HD and MB providing clinical expertise for featurisation. ER, KK, EM, NF and HD will conduct the data analyses. ER and EM wrote the first draft of the manuscript, all authors revised and reviewed this and approved the final manuscript.

**Funding** This work was supported by the UCL Rosetrees Stoneygate prize (M712), a Cystic Fibrosis Trust Clinical Excellence and Innovation Award (CEA010), A UCL Partners award and the HEFCE Higher Education Innovation Fund (KEI2017-01-04). GD is supported by a Wellcome Institutional Strategic Support Fund award at UCL (204841/Z/16/Z), and formerly an NIHR Clinical Trials Fellowship. HD is funded by the CF Trust Youth Activity Unlimited SRC and an NIHR GOSH BRC internship. All work at UCL GOSICH is supported by the NIHR GOSH BRC. The views expressed are those of the authors and not necessarily those of the NHS, the NIHR or the Department of Health. The study is sponsored by UCL. The funders and sponsor played no role in the design of the study.

**Competing interests** GD reports personal lecture fees from Chiesi Limited for an invited talk on Project Fizzyo at an educational event. All other authors report no conflicts of interest in relation to this protocol.

**Patient and public involvement** Patients and/or the public were involved in the design, or conduct, or reporting, or dissemination plans of this research. Refer to the 'Methods' section for further details.

**Patient consent for publication** Not required.

**Provenance and peer review** Not commissioned; externally peer reviewed.

**ORCID iDs**
Emma Raywood http://orcid.org/0000-0002-0993-5115
Gwyneth Davies http://orcid.org/0000-0001-7937-2728
Tempest Van Schaik http://orcid.org/0000-0001-7745-2249
Eleanor Main http://orcid.org/0000-0002-9739-3167

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
