## [Reviewer comments · BMJ Open]

ARTICLE DETAILS

TITLE (PROVISIONAL)	Protocol for Project Fizzyo, an analytic longitudinal observational cohort study of physiotherapy for children and young people with cystic fibrosis, with interrupted time series design
AUTHORS	Raywood, Emma; Douglas, Helen; Kapoor, Kunal; Filipow, Nicole; Murray, Nicky; O'Connor, Rachel; Stott, Lee; Saul, Greg; Kuzhagaliyev, Tim; Davies, Gwyneth; Liakhovich, Olga; Van Schaik, Tempest; Furtuna, Bianca; Booth, John; Shannon, Harriet; Bryon, Mandy; Main, Eleanor

VERSION 1 – REVIEW

REVIEWER	Mark Elkins SLHD, Australia
REVIEW RETURNED	21-May-2020

GENERAL COMMENTS	I acknowledge the strengths of the study as outlined in the paper. I believe it will produce an extremely valuable dataset. However, the "Strengths and limitations of the study" section lists only strengths and does not recognise the limitations inherent in the study protocol, such as confounding by the non-random allocation of the interventions, and confounding by order effects. My understanding is that the proposed analyses cannot fully account for these issues and therefore this should be recognised as a limitation of the study. I don't see the value in publishing a protocol for a study that has already commenced, especially since the registered protocol is already publicly available on ISRCTN. Because they are retrospective, neither of these two protocols will document whether the researchers have adhered to their original protocol and reported all outcomes as originally planned. Couldn't CYPwCF who use ACT less than once per day (or an completely non-compliant) to observe whether they take up ACT in order to engage with the software gaming? If ACT are effective, don't these CYPwCF have the most to gain? In addition to Acapella, it would be helpful to see how the sensor connects to the other ACT devices The validity of the main analysis will hinge upon the validity of the adherence data. Will the authors be able to identify the all elements of therapy (pressure, breath length, breaths per set, numbers of sets, duration, sessions per day) prescribed exactly in the patient record for each CYPwCF at all sites, with adjustments to the prescribed regimen due to fluctuations in clinical status also documented?
--

	Figure 1 legend: Delete "for". Remote monitoring, Activity tracker, paragraph 2, final sentence: Delete "to".
--	---

REVIEWER	Dr Zoe Saynor School of Sport, Health and Exercise Science, Faculty of Science and Health, University of Portsmouth, England, UK
REVIEW RETURNED	29-May-2020

GENERAL COMMENTS	Summary This manuscript provides a clear and comprehensive overview of the protocol for Project Fizzyo, which is a novel and clinically important study underway in children and adolescents with cystic fibrosis. Given that this is a protocol paper, my comments are largely requesting several additional details be added in sections. On the whole, this article is written to a high standard, with figures presented well and complementing the article nicely. More detailed comments / suggestions are provided below: Title: An appropriate title has been included. Perhaps it would be more appropriate to state children and adolescents with CF rather than 'young people', given that this may indicate young adults are also being included. Abstract: Suggest reword: 'The protocol for Project Fizzyo, which is using..' In line with the title, consider stating children and adolescents with CF rather than 'young people', given that this may indicate young adults are also being included. You are writing in the past tenses, however this is an ongoing study. Can you confirm that all participants have already been recruited? Please can you also provide details of the degree of pulmonary dysfunction that you were seeking to recruit for the study. Article summary: The authors have provided a nice a nice summary of some of the key strengths of this ongoing research study, however no limitations are offered. Please could you include a brief statement outlining these, for example this could include the choice of activity tracker. Introduction: Generally, the introduction is written to a high standard and provides a clear introduction and rationale for the present study, outlining contemporary and relevant literature. A few minor typographical recommendations: Page 6 Line 6: Suggest could add a reference where you state 'recent treatment advances' Page 6 Line 15: Comma not needed 'footstep' data is not a standard term within physical activity research, perhaps step count may be a better term to use here?
--

	Methods: Page 7 Line 35: Are you able to include any key modifications that were generated through this consultation with people with CF and their families? Page 7: I assume that assent rather than consent was obtained from minors? As such, please could you clarify this. Please can you outline and include details regarding how you standardised instructions and procedures between the different sites involved in the study? Page 8 Line 9: You use the CWPwCF abbreviation here but do not use it elsewhere Please can you provide further detail regarding how regularly you are collecting HR data and how you are analysing the physical activity data.
--	---

VERSION 1 – AUTHOR RESPONSE

Reviewer 1’s comments:

- I acknowledge the strengths of the study as outlined in the paper. I believe it will produce an extremely valuable dataset. However, the "Strengths and limitations of the study" section lists only strengths and does not recognise the limitations inherent in the study protocol, such as confounding by the non-random allocation of the interventions, and confounding by order effects. My understanding is that the proposed analyses cannot fully account for these issues and therefore this should be recognised as a limitation of the study.

- “Strengths and limitations” section amended as suggested (by both reviewers) to include limitations.
- We acknowledge the limitations of our study, and have now included these in relation to tracker choice and remote monitoring (3rd, 4th points). We believe that the order of interventions (feedback and gaming) on adherence in this study are not vulnerable to order effects and these are assessed carefully within the interrupted time series design. This study design was specifically chosen as an alternative to a randomised controlled trial (RCT), in part because previous RCTs for physiotherapy have failed to provide clear evidence. Allocation concealment is largely not possible in studies involving airway clearance techniques and this has been a well-recognised limitation of RCT designs for such studies. Previous studies have shown high dropout rates from less preferred interventions following randomisation. Furthermore, due to the personalisation of physiotherapy devices and techniques, heterogeneity of CF clinical profiles in children and the insensitivity of clinical end points such as FEV1 in this population, would make comparison of difference between groups within an RCT difficult.

- I don't see the value in publishing a protocol for a study that has already commenced, especially since the registered protocol is already publicly available on ISRCTN. Because they are retrospective, neither of these two protocols will document whether the researchers have adhered to their original protocol and reported all outcomes as originally planned.

- Our manuscript includes important data that is beyond the scope of ISRCTN, in line with BMJ Open guidelines which themselves state that “Publishing protocols in full also makes available more information than is currently required by trial registries” (<https://bmjopen.bmj.com/pages/authors/#protocol>). The bespoke ACT monitoring system and data pipeline developed for this study may have a wider application and we think there is value in providing more details than are currently available in the ISRCTN protocol.
- Although the study is in progress, the data collection is not complete. This again is in-line with the BMJ Open publishing guidelines for protocols.

- Couldn't CYPwCF who use ACT less than once per day (or an completely non-compliant) to observe whether they take up ACT in order to engage with the software gaming? If ACT are effective, don't these CYPwCF have the most to gain?

- We have updated the text to reflect that to be recruited children did not have to be adherent to ACT (page 5, sentence added to second paragraph of "Participants" section and replace word "using" for "prescribed" in this section).

- To clarify; participants must be prescribed a study appropriate ACT one or more times daily. If they do not use it (are self-reported non-adherent) this did not exclude them from participation. We agree that these children may be the ones who stand to gain the most in terms of improved adherence and are an important group to include.

- In addition to Acapella, it would be helpful to see how the sensor connects to the other ACT devices

- Figure 3 updated to show the sensor and other compatible devices, caption and reference in text to figure also updated.

-The validity of the main analysis will hinge upon the validity of the adherence data. Will the authors be able to identify the all elements of therapy (pressure, breath length, breaths per set, numbers of sets, duration, sessions per day) prescribed exactly in the patient record for each CYPwCF at all sites, with adjustments to the prescribed regimen due to fluctuations in clinical status also documented?

- This is our intention, as a key part of the guidelines for physiotherapy prescription is that it is personalised.

- The main features as stated should be listed in clinical physiotherapy notes (from clinic visits, annual review, admissions etc) at each site, and this will be included in our analysis: "including documented changes to technique" added to clarify on page 8 "clinical data" section. We also ask participants about their prescription at 3 time points (recruitment, mid-point, end of study) at which times we hope to capture any changes. Participants are able to inform the study team if prescription changes (e.g. during the feedback stage this may mean their "target" number of daily breaths is incorrect), but this is not expected of them We appreciate that this record may not always be accurate but do not wish to increase burden of participation.

-Figure 1 legend: Delete "for".

- Word deleted

- Remote monitoring, Activity tracker, paragraph 2, final sentence: Delete "to".

- Repeat word deleted

Reviewer 2's comments:

Title:

-An appropriate title has been included. Perhaps it would be more appropriate to state children and adolescents with CF rather than 'young people', given that this may indicate young adults are also being included.

- We have left this unchanged to remain consistent the title as registered with ISRCTN however acknowledge it is worth considering for future publications.

Abstract:

-Suggest reword: 'The protocol for Project Fizzyo, which is using..'

- Amended to be "which uses"

-In line with the title, consider stating children and adolescents with CF rather than 'young people', given that this may indicate young adults are also being included.

- See above comment

-You are writing in the past tenses, however this is an ongoing study. Can you confirm that all participants have already been recruited?

- Yes all participants are recruited but data collection is ongoing, methods and analysis section of abstract rephrased to reflect this.

-Please can you also provide details of the degree of pulmonary dysfunction that you were seeking to recruit for the study.

- No change made to text of abstract (due to word count restriction).
- To confirm: the inclusion criteria did not specify FEV1 or other demographic characteristics (page 5). If the clinical team felt it was appropriate for the patient to be approached a low lung function value was not a reason for exclusion.

Article summary:

- The authors have provided a nice a nice summary of some of the key strengths of this ongoing research study, however no limitations are offered. Please could you include a brief statement outlining these, for example this could include the choice of activity tracker.

- “Strengths and limitations” section amended as suggested (by both reviewers) to include limitations.
- Limitations related to tracker choice and remote monitoring included (3rd, 4th points)

Introduction:

-Page 6 Line 6: Suggest could add a reference where you state ‘recent treatment advances’

- Reference included and sentence to indicate this refers to modulator therapies.
- All references renumbered

-Page 6 Line 15: Comma not needed

- Removed comma

-‘footstep’ data is not a standard term within physical activity research, perhaps step count may be a better term to use here?

- We agree step count is better than footsteps (amended throughout document, 6 replacements)

Methods:

- Page 7 Line 35: Are you able to include any key modifications that were generated through this consultation with people with CF and their families?

- More details are in the newly included PPI section on page 4 (see formatting amendments below).

- Page 7: I assume that assent rather than consent was obtained from minors? As such, please could you clarify this.

- Sorry if unclear, wording changed. Child assent and parental consent was sought from all participants. If participants were over the legal age of consent (16y) and judged to be competent by their clinical team they were able to consent for themselves (though always in the presence of a parent) as recommended at our NHS REC review.

- Please can you outline and include details regarding how you standardised instructions and procedures between the different sites involved in the study?

- We have added extra detail to the “data collection” section on pages 5 and 6 regarding the participant instructions and standardised order of assessments.

-Page 8 Line 9: You use the CWPwCF abbreviation here but do not use it elsewhere

- I cannot find an instance of CWPwCF for correction. CYPwCF unchanged as is defined on page 3 and used elsewhere.

-Please can you provide further detail regarding how regularly you are collecting HR data and how you are analysing the physical activity data.

- Sentence added to the section “Remote monitoring, Activity tracker” (page 7). To indicate that HR data is collected continuously during Fitbit wear for 16 months. The next sentence indicates participants are told to wear the Fitbit throughout waking hours and charging instructions.
- Sentence added to the final paragraph of section “Data analysis, Adherence to physiotherapy” (page 9) to add detail to MVPA threshold calculation and use of step count.

VERSION 2 – REVIEW

REVIEWER	Mark Elkins Sydney Local Health District Australia
REVIEW RETURNED	16-Jul-2020

GENERAL COMMENTS	Nil
-----

REVIEWER	Dr Zoe Saynor School of Sport, Health and Exercise Science Faculty of Science and Health University of Portsmouth Portsmouth UK
REVIEW RETURNED	16-Jul-2020

GENERAL COMMENTS	The authors have satisfactory addressed all of the reviewer comments.
---